# Purification Effect of Sequential Constructed Wetland for the Polluted Water in Urban River

**Xueyuan Bai [1], Xianfang Zhu [2], Haibo Jiang [1], Zhongqiang Wang [1]**  **, Chunguang He [1,*], Lianxi Sheng [1,*] and Jie Zhuang [3]**

[1] State Environmental Protection Key Laboratory of Wetland Ecology and Vegetation Restoration, Key Laboratory for Vegetation Ecology, Ministry of Education, Jilin Provincial Key Laboratory of Ecological Restoration and Ecosystem Management, Northeast Normal University, Changchun 130117, China; baixy324@nenu.edu.cn (X.B.); jianghb625@nenu.edu.cn (H.J.); wangzq027@nenu.edu.cn (Z.W.)

[2] Key Laboratory of Water and Sediment Sciences, Ministry of Education, Department of Environmental Engineering, Peking University, Beijing 100871, China; xianfangzhu@pku.edu.cn

[3] Department of Biosystems Engineering and Soil Science, University Tennessee, Knoxville, TN 37996, USA; jzhuang@utk.edu

* Correspondence: he-cg@nenu.edu.cn (C.H.); shenglx@nenu.edu.cn (L.S.); Tel.: +86-431-891-65611

**Abstract:** Constructed wetlands can play an active role in improving the water quality of urban rivers. In this study, a sequential series system of the floating-bed constructed wetland (FBCW), horizontal subsurface flow constructed wetland (HSFCW), and surface flow constructed wetland (SFCW) were constructed for the urban river treatment in the cold regions of North China, which gave full play to the combined advantages. In the Yitong River, the designed capacity and the hydraulic loading of the system was 100 $m^3$/d and 0.10 $m^3/m^2$d, respectively. The hydraulic retention time was approximately 72 h. The monitoring results, from April to October in 2016, showed the multiple wetland ecosystem could effectively remove chemical oxygen demand (COD), ammonia nitrogen ($NH_4^+$-N), total nitrogen (TN), total phosphate (TP), and suspended solids (SS) at average removal rates of 74.79%, 80.90%, 71.12%, 78.44%, and 91.90%, respectively. The removal rate of SS in floating-bed wetland was the largest among all the indicators (80.24%), which could prevent the block of sub-surface flow wetland effectively. The sub-surface flow wetland could remove the $NH_4$-N, TN, and TP effectively, and the contribution rates were 79.20%, 64.64%, and 81.71%, respectively. The surface flow wetland could further purify the TN and the removal rate of TN could reach 23%. The total investment of this ecological engineering was $12,000. The construction cost and the operation cost were $120 and $0.02 per ton of polluted water, which was about 1/3 to 1/5 and 1/6 to 1/3 of the conventional sewage treatment, respectively. The results of this study provide a technical demonstration of the restoration of polluted water in urban rivers in northern China.

**Keywords:** polluted urban river; sequential constructed wetlands; purification effect; water restoration; Yitong River

## 1. Introduction

In recent decades, with the rapid development of urban economies and urbanization, urban rivers all over the world have been facing the threat of pollution. Japan, the United States, and some European countries began to implement water purification of urban rivers in the 1950s and 1960s [1]. However, the research and application of further technology was only carried out in the past 20 to 30 years [2,3]. At present, river water purification technologies can be categorized into physical, chemical, biological, and ecological methods. Physical methods include aeration [4] and sediment

dredging [5–7]. Chemical methods include chemical precipitation [8] and the application of chemical algaecide [9]. Biological methods include bioremediation [10], biofilms [11,12], contact oxidation [13–15], and membrane bioreactor technology [16,17]. Ecological methods include ecological ponds [18,19], plant purification treatment [20,21], ecological floating beds [22–24], and constructed wetlands [25–27]. Among these methods, aeration is the easiest, fastest, most effective, and most widely applied. It has been applied to the restoration of the Emscher River in Germany, the Homewood Canal in the United States, and the Thames River in the United Kingdom [28]. However, the operation and maintenance costs are high. Sediment dredging can remove endogenous pollution sources and is also a widely applied method to improve the water environment. There are problems associated with this method, such as the large amount of construction work, challenges to processing the sludge, and the additional pollution created by improper dredging [29,30]. Physical and chemical methods are often used only as secondary approaches to mediate pollution or to treat emergent water pollution. Although biological methods effectively clean up pollution and have low energy consumption requirements and small environmental impacts, they also have issues such as time-consuming processes to cultivate microorganisms and purification that is subject to restrictions from external conditions such as the temperature and water flow. Biofilm technology requires large-scale construction and involves land use issues. Constructed wetlands, on the other hand, use physical, chemical, and biological synergies in an ecosystem to efficiently remove pollutants by simulating the natural environment [31]. The operating costs are low, and they are easy to maintain and manage without secondary pollution. They also offer ecological benefits: they have been demonstrated to be an economical and efficient sewage treatment and management method [32], and they have become a preferred ecological method to improve the water quality of rivers in cities around the world. Therefore, the development of optimized functions of constructed wetlands that suit different goals and requirements has been a focal area of research. However, the ecological ponds occupy a large area of land, which is easily affected by environmental conditions, and the removal effect of nitrogen and phosphorus is not stable. The plant purification treatment and ecological floating beds only rely on aquatic plants to absorb nitrogen, phosphorus, and organic matter; thus, the capacity is limited. The constructed wetlands have disadvantages for purifying water because of the large area and low hydraulic load for the surface flow constructed wetland (SFCW); moreover, the horizontal subsurface flow constructed wetland (HSFCW) has a poor resistance to shock loads and is easily blocked. Therefore, the river water purification is a complex system engineering, and needs the joint application of various ecological technologies. The Yitong River is the mother river of Changchun, a capital city in northeast China. To create a water landscape for the city, multiple dams have been constructed to reserve water on the river section within Changchun. As a result, the flow rate has become slow, the dissolved oxygen (DO) level has decreased sharply, the river's self-purification capacity has been lost, and the water quality has begun to deteriorate. Eutrophication is especially severe in seasons with high temperatures. This situation has brought hidden risks to the water environment and the ecological security of the areas of Changchun near the lower reaches of the river, which severely affects the surrounding ecological environment and the daily life of nearby residents and restricts the sustainable development of the urban waters.

In this paper, we built a sequential combination system of the floating-bed constructed wetland (FBCW) + horizontal subsurface flow constructed wetland (HSFCW) + surface flow constructed wetland (SFCW) to purify the polluted water of Yitong River, using the favorable terrain of the park on the shore. This system was continuously monitored for one year and the results were analyzed. They can provide necessary data support for the popularization and application of the technology.

## 2. Method

### 2.1. Study Area

The study area is along the Yitong River in the city of Changchun, Jilin Province, northeast China (Figure 1, 43°50′32″ N, 125°21′32″ E), which has a temperate continental monsoon climate zone.

The annual rainfall in the study area is 600 to 700 mm. The average annual temperature is 4.6 °C. The maximum temperature in the summer is 40 °C. The annual freezing period is 5 months.

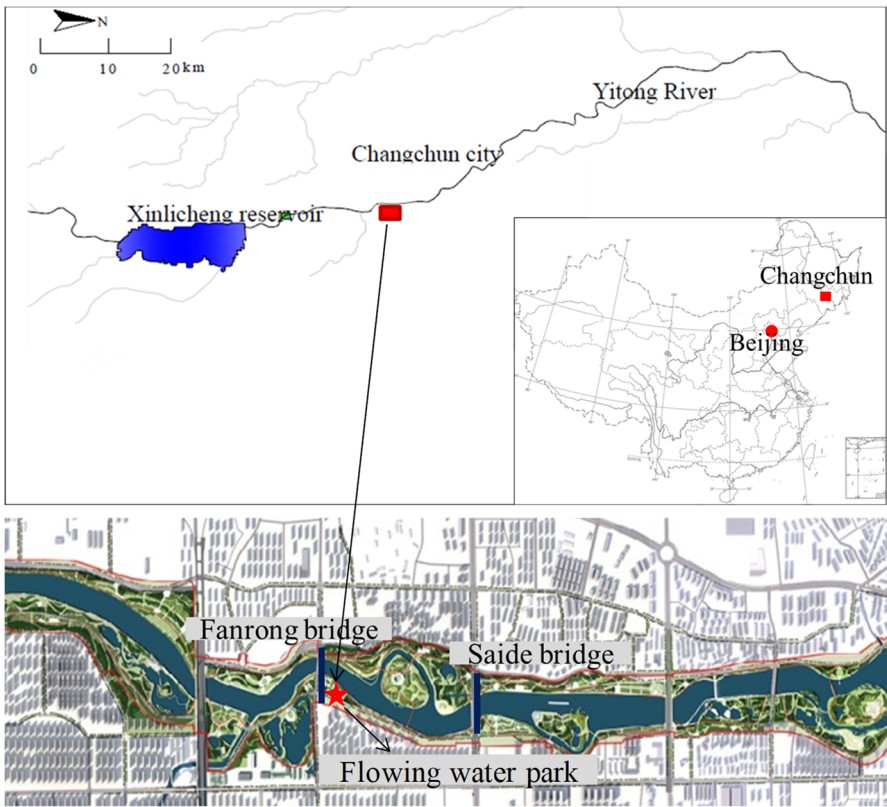

**Figure 1.** Location and overview of the study area.

The water quality was monitored in the study area from April to October 2014. Water quality data are shown as averages ± standard deviations (Table 1). The average chemical oxygen demand (COD) and ammonia nitrogen ($NH_4^+$-N) concentrations were 57.65 mg/L and 5.60 mg/L, respectively, which were both beyond the lower limits of Class V water in the Environmental Quality Standards for Surface Water (GB3838-2002) [33]. Therefore, the water quality was classified as worse than Class V.

**Table 1.** Results of water quality monitoring in the study area of the Yitong River.

| Water Quality Indicators | Range (mg/L) | Mean |
|:---:|:---:|:---:|
| pH | 6.89–7.28 | 7.08 ± 0.13 |
| COD | 45.48–63.46 | 57.65 ± 6.28 |
| $NH_4^+$-N | 4.65–6.42 | 5.60 ± 0.68 |
| TN | 5.54–7.39 | 6.57 ± 0.71 |
| TP | 1.43–2.24 | 1.81 ± 0.31 |
| SS | 40.26–62.37 | 54.04 ± 8.04 |
| DO | 1.32–2.16 | 1.69 ± 0.28 |

COD—chemical oxygen demand; $NH_4^+$-N—ammonia nitrogen; TN—total nitrogen; TP—total phosphate; SS—suspended solids; DO—dissolved oxygen.

### 2.2. Ecological Engineering Design

The engineering process is shown in Figure 2. The wetland system was composed of FBCW, HSFCW, and SFCW in sequence. Firstly, the water of the Yitong River was introduced into FBCW by gravity from the upstream rubber. The FBCW mainly realized the pre-treatment function, introducing the adsorption carrier and suspended biological filler, which attached a large number of microbial flora,

and formed a collaborative purification system of aquatic plants, porous substrate, and biofilm. This is to improve the degradation efficiency of organics, nitrogen, and phosphorus, reduce the pollution load for follow-up HSFCW, and improve the impact resistance of the system. Additionally, the FBCW can remove suspended solids (SS) effectively and reduce the risk of clogging of the HSFCW. The FBCW used the terrain elevation difference to design a multi-level water drop for reaeration, and created favorable conditions of the denitrification for HSFCW. The HSFCW was the core of the system, and formed the aerobic-anoxic-anaerobic environment, which contributed to the smooth progress of nitrification and denitrification, and thus, could effectively promote the removal of nitrogen in sewage. Falling water and reoxygenation by FBCW improved the concentration of DO, which was conducive to the transformation of $NH_4^+$-N and improved the nitrification reaction capacity in the front end of HSFCW. However, in the middle and back end of HSFCW, the anoxic and anaerobic environment, which for the DO was lower, could promote the denitrification process. Then, the effluent from the HSFCW entered the SFCW. The SFCW was rich in microtopography, which could improve the DO, further removed the pollutants of nitrogen and phosphorus, and provided a safety guarantee for the effluent reaching the standard. The effluent of SFCW flowed back to Yitong River. The system has played an effective collaborative role of FBCW, HSFCW, and SFCW, which reduced the risk of wetland blocking and improved the antipollution load capacity and operation stability. At last, the system improved the efficiency of pollutant purification.

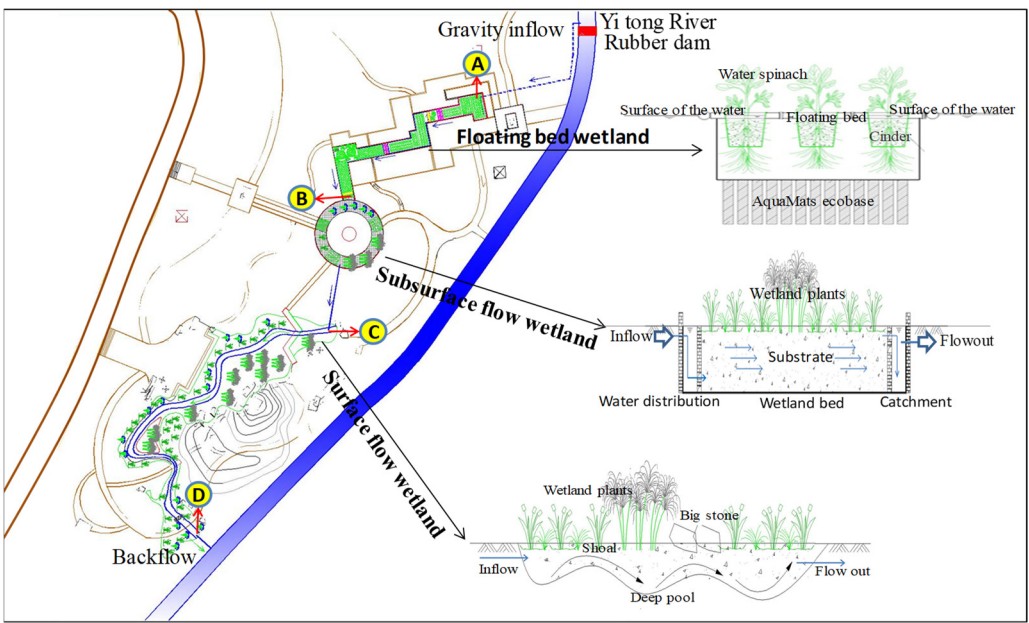

**Figure 2.** Processing flow and configuration.

The designed water volume was 100 m$^3$/d. According to the water quality monitoring results in 2014, the design of influent water was COD ≤ 70 mg/L, $NH_4^+$-N ≤ 8.0 mg/L, SS ≤ 60 mg/L. Therefore, the design of effluent water was COD ≤ 40 mg/L, $NH_4^+$-N ≤ 2.0 mg/L, SS ≤ 6.0 mg/L. The total surface hydraulic loading was approximately 0.10 m$^3$/m$^2$d. The total area of the wetlands was 1000 m$^2$. The hydraulic retention time was 3 d. Among them, the surface hydraulic loading of FBCW, HSFCW, and SFCW was 0.42 m$^3$/m$^2$d, 0.50 m$^3$/m$^2$d, and 0.18 m$^3$/m$^2$d, respectively. The area of FBCW, HSFCW, and SFCW was 240 m$^2$, 200 m$^2$, and 560 m$^2$, respectively. The hydraulic retention time of FBCW, HSFCW, and SFCW was 0.83 d, 1.17 d, and 1.0 d, respectively.

The FBCW comprised 25 floating beds, with 3.00 × 2.00 m per floating bed. The total area of floating beds was 150 m$^2$, and a coverage rate was 62.5% in FBCW. The top of the floating beds was set up with 3000 planters with diameters of 10cm and heights of 30 cm (20 planters/m$^2$). They were filled with absorbent volcanic rocks as the substrate to a height of 25 cm. Two water spinach plants were planted in

every planter. AquaMats (Hangzhou Zijing Envrionmental Engineering CO.,LTD., Hangzhou, China), a biofiltration media, were hung 20 cm apart from one another under the floating beds. The height of the media was 30 cm.

The depth of the HSFCW was 1.2 m. The SSFCW was filled with natural volcanic rocks with a particle size of 10–20 cm and a filling height of 1.0 m. *Lythrum salicaria L.*, a wetland plant, was planted in a density of 25 clumps/m$^2$ and three plants/clump. The distribution zone and the catchment zone were established front and after end in the wetland to ensure the even distribution of water in the system and to maximize the wetland efficiency.

The designed depth of SFCW was 0.1 m to 0.8 m. The SFCW was comprised of shallow waters, deep pools, sandbars, and ecological islands, with *Lythrum salicaria L.*, *Iris pseudacorus L.*, *Salix integra*, and other wetland plants. It was also supplemented by ecological bag revetment and slope bank with vegetation.

*2.3. Data Collection and Analysis*

The water quality was continuously monitored from April to October 2016. Samples were taken from 9:00–11:00 on the 17th of each month. Sampling staff used clean plastic samplers to take water samples at the inflow and outflow of each wetland along the direction of the flow. The sampling points are shown in Figure 2: sampling point A (system inflow), sampling point B (outflow of the FBCW), sampling point C (outflow of the HSFCW), and sampling point D (outflow of SFCW). Three samples of 500 mL water were taken in clean plastic sampling bottles at each sampling point as duplicate samples. A water quality analysis of the samples was conducted within 24 h. The main monitoring indicators included pH, SS, DO, COD, NH$_4^+$-N, total nitrogen (TN) and total phosphate (TP). The methods were based on the Water and Wastewater Monitoring and Analysis Method (Fourth Edition) [34].

## 3. Results

The system was operated from April to June in 2016. The effectiveness of the removal of various pollutants gradually stabilized, and the water quality of the system improved significantly. Monitoring data showed that the SS concentration of inflow fluctuated between 46.38 and 89.26 mg/L (the mean was 64.84 mg/L) while the system was operating. The SS concentration of the outflow was 4.35–5.68 mg/L (the mean was 4.90 mg/L). The removal rate of SS was 87.86%–94.80% (the mean was 91.90%). From June to September, the root system of the plants in the floating-bed wetland continued to grow. After the formation of a stable biofilm on the hanging biofiltration media, the filtration by the plant roots and the interception and adsorption of the biofiltration media became prominent. The maximum removal rate of SS was 94.80%. As seen in Figure 3, the average removal rate of SS by the FBCW was 80.24%. The removal of SS in the HSFCW increased by 9.45% on average compared to the outflow of the FBCW. The removal of SS in the SFCW increased by 2.21% on average compared to the outflow of HSFCW. These results indicated that the removal of SS was mainly completed in the FBCW. The HSFCW and the SFCW enhanced the removal of SS.

During the operation of the system, the DO concentration of the inflow was basically maintained at 1.43–1.86 mg/L. The average DO concentration of the system inflow was 1.59 mg/L. The DO concentration of the outflow was 3.86–5.58 mg/L. The average DO concentration of the outflow was 4.79 mg/L. The DO concentration of the outflow met the water quality standard of Class IV water in the Environmental Quality Standards for Surface Water (GB3838-2002) [33]. As seen in Figure 4, oxygenation by the waterfalls in the FBCW significantly increased the DO concentration of the outflow. The consumption of the HSFCW brought the DO concentration back down to a lower level at the outflow of the HSFCW. Then, the effect of the SFCW gradually improved the DO concentration of the system outflow. The DO concentration in general showed a pattern of first increasing, then decreasing, and increasing again. The DO concentration of the outflow increased by an average of 2.03 times. The DO concentration of the outflow reached a maximum of 5.58 mg/L. The increase in DO was mainly

attributed to the waterfalls in the FBCW. The terrain changes in the SFCW, such as deep ponds, shallow waters, jumps, and waterfalls, adjusted the form and speed of the flow.

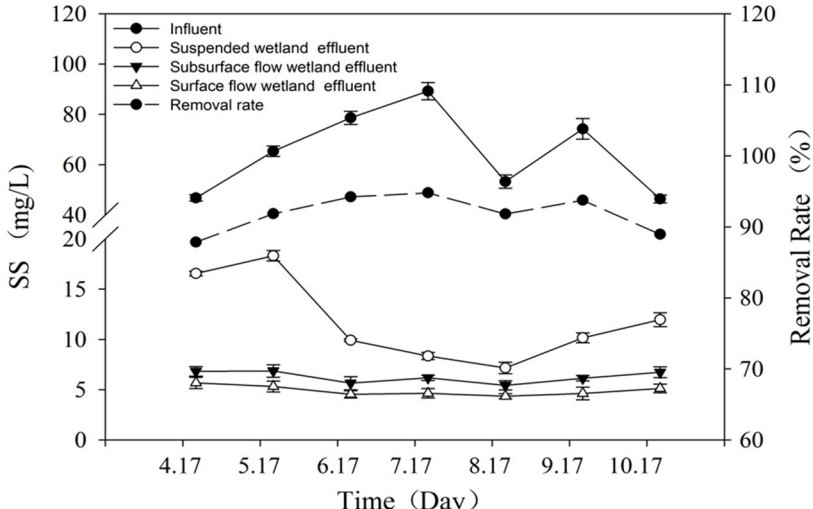

**Figure 3.** Concentration of SS in the inflow and outflow and the removal rate.

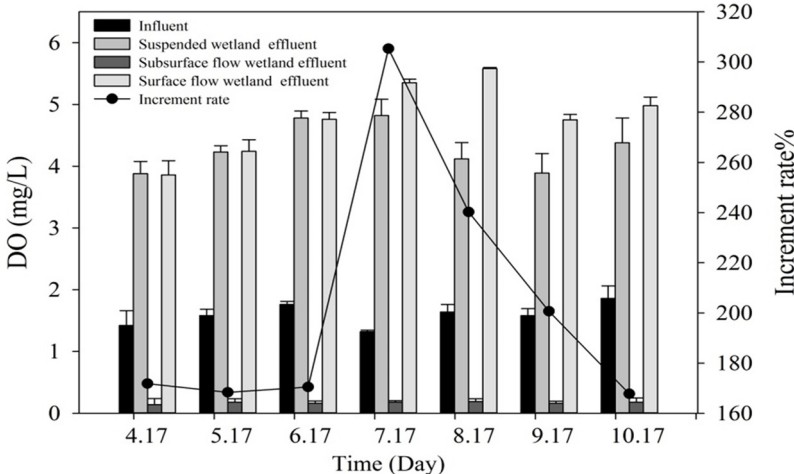

**Figure 4.** DO concentration of the inflow and the increase rate.

During the operation of the system, the COD concentration of the inflow fluctuated between 62.35 and 83.48 mg/L. The average COD concentration of the system inflow was 71.57 mg/L. The average COD concentration of the outflow was 20.57 mg/L. The COD removal rate of the system was 38.14%–82.05%, and the average removal rate was 74.79%. As seen in Figure 5, the COD removal rate was the lowest in April at 38.14%. It rose to 61.29% in May. With the exception of April, May, and October, when the system just started and reached the end of operation, the COD removal rate could reach more than 80%. The COD concentration of the outflow in June to September met the water quality standard of Class III water in the Environmental Quality Standards for Surface Water (GB3838-2002) [33]. In this system, the contribution of the HSFCW to the COD removal was close to 54.42%. The floating-bed wetland contributed approximately 34.56% to the COD removal. The contribution of the SFCW was only approximately 11.02% because the COD removal by the HSFCW was mainly achieved through microbial degradation and adsorption by plants and filtration media. Compared to the COD removal rate in the HSFCW, the COD removal rate in the FBCW was lower due to the shorter retention time. Additionally, the SFCW was at the downstream end of the system where the COD concentration of

the inflow was lower. In addition, the SFCW had a limited ability to remove pollutants thus, its COD removal rate was not high.

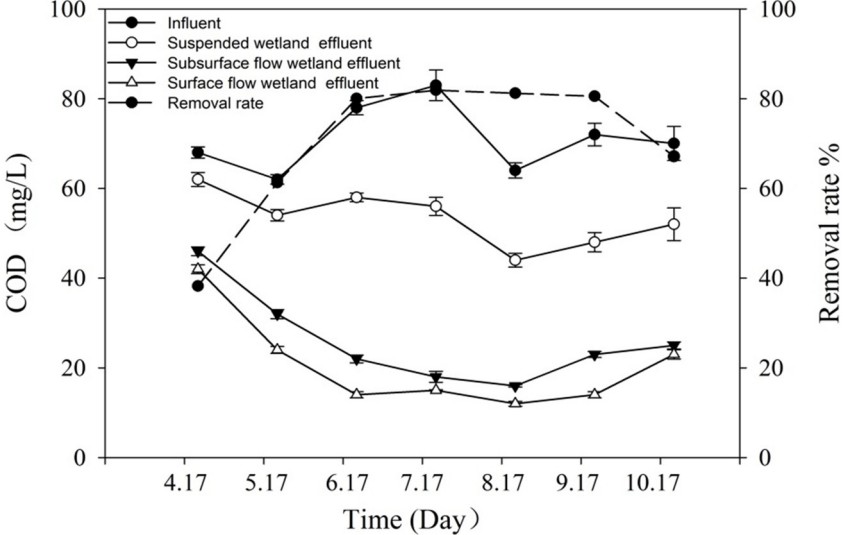

**Figure 5.** COD concentration of the inflow and outflow and the removal rate.

During the operation of the system, the $NH_4^+$-N concentration of the inflow was 5.65–8.26 mg/L. The average $NH_4^+$-N concentration of the inflow was 6.98 mg/L. The average $NH_4^+$-N concentration of the outflow was 1.25–4.66 mg/L, and the average removal rate was 80.90%. As seen in Figure 6, excluding the initial stage in April and May and the late operation season in October, the average $NH_4^+$-N concentration of the outflow from June to September was 1.35 mg/L. The removal rate of $NH_4^+$-N was higher than 78%. The $NH_4^+$-N concentration of the outflow met the water quality standard of Class IV water in the Environmental Quality Standards for Surface Water (GB3838-2002) [33]. The removal of $NH_4^+$-N in the entire wetland system was largely achieved in the HSFCW, which contributed to 79.20% in the system purification.

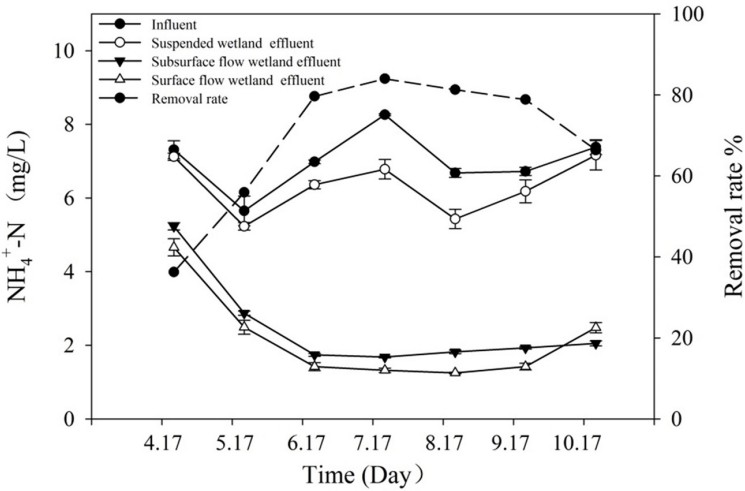

**Figure 6.** Concentration of $NH_4^+$-Nin the inflow and outflow and the removal rate.

During the operation of the system, the concentration of TN in the inflow was 6.43–8.89 mg/L. The average TN concentration of the system inflow was 7.73 mg/L. The TN concentration in the outflow was 1.75–6.28 mg/L, and the average removal rate was 71.12%. As seen in Figure 7, in April, the TN concentration in the outflow was 6.28 mg/L, and the removal rate was only 19.00%. In May, the TN

concentration in the outflow was 4.16 mg/L, and the removal rate was approximately 35%. From June to September, the average TN concentration of the outflow was 1.83 mg/L, and the removal rate reached approximately 76.75% and was stable. In October, the TN concentration in the outflow was 2.64 mg/L, and the removal rate decreased. The TN removal rate by the FBCW, the HSFCW, and the SFCW was 12.35%, 64.66%, and 23.00%, respectively

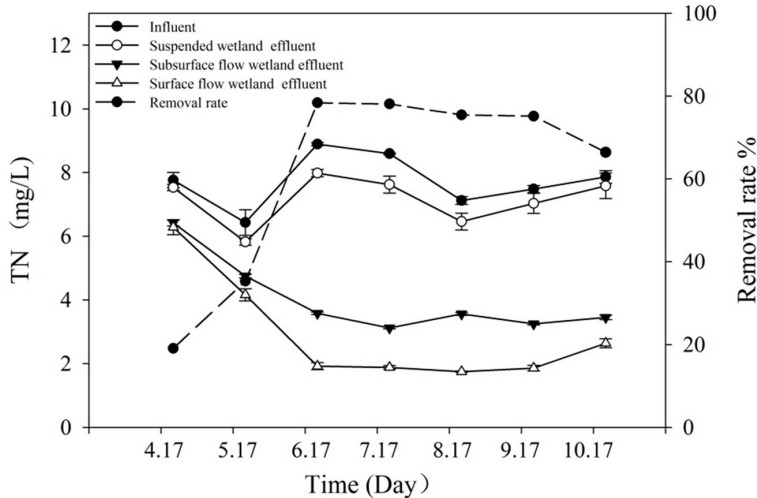

**Figure 7.** Concentration of TN in the inflow and outflow and the removal rate.

During the operation of the system, the concentration of TP in the inflow was 1.48–1.69 mg/L. The average TP concentration in the inflow was 1.56 mg/L. The TP concentration in the outflow was 0.32–0.38 mg/L. The average TP concentration in the outflow was 0.35 mg/L. As seen in Figure 8, the removal rate of TP reached an average of 78.39%. The TP concentration in the outflow met the water quality standard of Class V water in the Environmental Quality Standards for Surface Water (GB3838-2002) [33]. The system had a strong removal capacity for TP, which was mainly attributed to the HSFCW. The removal of TP by the FBCW and the SFCW was not significant, with an average removal rate of 12.86% and 5.43%, respectively. This might be related to the form of phosphorus and the mechanism and method of removal. It is generally believed that the removal of phosphorus in artificial wetlands is mainly achieved by adsorption to the substrates. Therefore, it is possible that the substrate structures of the HSFCW played a major role in the adsorption and retention of phosphorus. The FBCW and SFCW only removed a small portion of the suspended phosphorus.

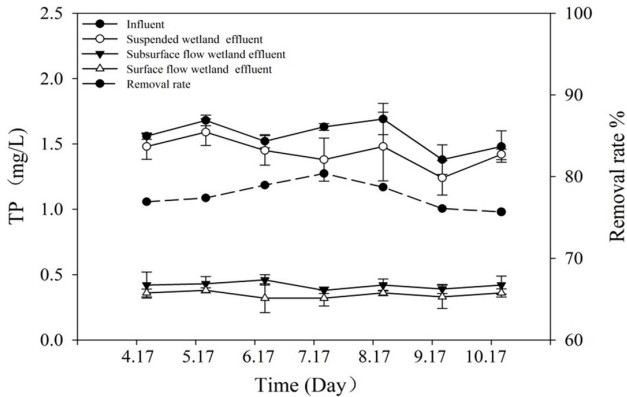

**Figure 8.** Concentration of TP in the inflow and outflow and the removal rate.

## 4. Discussion

Due to the cold winter in northern China, artificial wetlands cannot function properly in the winter. As a result, the wetland system stops operating in November every year. It restarts in March to April the next year when the weather becomes warmer again. When the system first started operating, the low temperature inhibited microbial activities, and the plants were not growing yet. The system mainly relied on substrate adsorption and interception to purify the water, and the efficiency of pollutant removal was low. As shown in Figures 5–7, in April, with an average temperature of 3–12 °C, the removal rates of COD and $NH_4^+$-N were less than 40%, and the removal rate of TN was only 19.00%. In May, the removal rates of COD, $NH_4^+$-N and TN increased to 61.29%, 55.92%, and 35.30%, respectively, when the average temperature was 12–23 °C and the wetland plants started to grow. Between June to September, the temperature gradually rose and the average temperature was 16–29 °C, when the plants grew up. As the operating time increased, the operation of the system gradually became stable and the microbial activities increased significantly. The integrated effect of the wetland plants, substrate, and microorganisms was fully functional, thus, the purification efficiency was best. The both removal rates of COD and $NH_4^+$-N were over 80%, when the TN could exceed 75%. After September, the temperature gradually decreased, and the plants stopped growing and started to wither. When the average temperature was 2–4 °C in October, the removal rates of COD, $NH_4^+$-N, and TN decreased to 67.14%, 66.39%, and 66.41%, respectively. The average temperature was −7–2 °C in November, and the wetland system progressively stopped running. Existing studies have shown that the nitrogen removal in a wetland is achieved by microbial activities, substrate adsorption, and plant absorption [35,36]. Low temperatures seriously affect the efficiency of water purification in artificial wetlands [37,38], because denitrifying microorganisms in artificial wetlands are active only when the temperature is above 15 °C [39,40]. In addition, during the operation of the system from June to September, despite the poor water quality of the inflow, the wetland system still operated stably. The removal rates of COD and NH4+-N still exceeded 75%. The COD concentration of the outflow was ≤20 mg/L, and the NH4+-N concentration of the outflow was ≤1.5 mg/L. The system showed a good effectiveness for pollutant removal.

The purpose of having a combination of different types of wetlands is to balance the advantages and disadvantages of each type of wetland [41]. Such a design can break through the restrictions imposed by the technical characteristics and environmental conditions of a single wetland type, enhance the system's resistance to loading fluctuations, and improve the efficiency of the water treatment [42]. According to a survey of more than 100 constructed wetlands in the United States, nearly half of them became clogged within five years of operation [43]. It is believed that the accumulation of SS is the main cause of blockage in the constructed wetlands [44]. Studies have also shown that horizontal and vertical flow wetlands can remove more than 90% of organic matter and almost all N and P if effective pretreatment facilities are set up prior to the constructed wetlands or if the wetland area is large enough [45]. Compared to a settling basin or an oxidation pond, using a FBCW as a pretreatment unit for the HSFCW can greatly improve the removal rate of SS and the purification efficiency of COD, nitrogen, and phosphorus. It can reduce the pollution load for the subsequent wetland process, improve the resistance of the loading capacity of the entire wetland system, and reduce the risk of clogging the HSFCW. The HSFCW is the core of the multiple wetlands and the key to effective denitrification in the system. The nitrification and denitrification by the microorganisms are the important processes for denitrification in constructed wetlands [46]. Golterman found that there must be both aerobic and anaerobic conditions in a wetland to achieve the nitrification and denitrification processes and the effective removal of nitrogen [47]. The nitrification process is aerobic. $NH_4^+$-N is mainly removed by nitrification. The DO concentration should normally be maintained above 2.0 mg/L for nitrification. Although the FBCW was not efficient in removing $NH_4^+$-N, the waterfalls designed by using the height differences in the terrain enabled oxygenation. This maintained a higher concentration of DO before water went into the HSFCW and promoted nitrification in the upper stream of the HSFCW and the conversion and removal of $NH_4^+$-N in the HSFCW. Denitrification requires anaerobic environments.

The DO level in the middle and lower streams of the HSFCW decreased rapidly. A hypoxic or anaerobic environment was formed, which promoted the process of denitrification and achieved an ideal removal of TN from the system. The removal of TN by a single type of wetland is not effective, because it is difficult to form both aerobic and hypoxic environments [31]. Multiple wetlands can effectively improve the level of DO in the outflow through the FBCW and form an anaerobic environment in the middle and lower streams in the HSFCW, which are conducive to denitrification in the wetland and critical to the effective denitrification of the system. Compared to the HSFCW, the average removal rate of $NH_4^+$-N was much lower in the SFCW in the lower stream of the system because the concentration of $NH_4^+$-N was lower in the inflow. However, the removal rate of TN in the SFCW was significant, accounting for 23% of the total removal of TN in the wetland system. This might be because the SFCW was usually in a saturated state during the operation and had considerable oxygen consumption, which was conducive to denitrification. In addition, the DO concentration going into the SFCW was very low. The multiple changes in the terrain between deep pools and shallow waters helped form a local environment with aerobic-hypoxic-anaerobic microcirculations, which promoted denitrification and the removal of TN. In addition, the seasonal temperature differences are large in northern China, and seasonal changes have a great impact on the effectiveness of constructed wetlands [25]. Multiple constructed wetlands can effectively make up for the disadvantage of a single wetland type [41], prolong the operation time of wetlands, and improve the efficiency of wetland purification.

For improved sewage treatment methods such as cyclic-activated sludge technology, activated charcoal adsorption, biofilms, and contact oxidation, the construction cost is usually $300–600 per ton of sewage, and the operating cost is approximately $0.07–0.15 per ton of sewage. Not only are the construction and operation costs high, but the management is also complicated. The construction cost of a constructed wetland system is approximately $100–180 per ton of sewage. The operating cost is approximately $0.014–0.056 per ton of sewage. Compared with the traditional biochemical treatment, constructed wetlands have the advantages of low investment, low operating cost, simple operation and management, and high ecological efficiency [48]. Therefore, these systems have great advantages in the restoration of polluted rivers. In this study, the total investment in construction was only $0.12 million, which was $120 per ton of water, and was about 1/5–1/3 of the investment in the construction of traditional sewage treatment. The system used gravity flow, no power was needed for operation, and the operating cost only involved regular maintenance. The maintenance costs were covered by the park maintenance budget, and the park created a part-time maintenance job for the wetlands. The part-time staff's wage was 70 dollar/month, and the system maintenance costs were only 0.023 dollar per ton of water, equal to 1/6–1/3 of the traditional sewage treatment. Compared with generic constructed wetland systems, the system described in this paper has a lower operating cost. Currently, the economic factor has become an important parameter for the selection of sewage treatment methods. Compared with operating constructed wetlands and other river water purification technologies in China and other countries, the multiple wetland technology in this study has obvious advantages and practicality.

## 5. Conclusions

(1) The one-year continuous monitored results of the sequential wetland system of FBCW, HSFCW, and SFCW showed that the system had a good effect on pollution removal. The system was mature and stable between June and September, and the average removal rates of COD, $NH_4^+$-N, TN, TP, and SS were 74.79%, 80.09%, 71.12%, 78.39%, and 91.90%, respectively. The FBCW had the highest removal rate of SS (80.24%), which could effectively prevent the blocking of HSFCW. The HSFCW contributed 79.20%, 64.64%, and 81.71% to the total removal of $NH_4^+$-N, TN, and TP, respectively. However, the SFCW could further remove TN, which contributed 23.00% to the total TN. The construction cost of the system was only $120 per ton of water, and the operation cost was only 0.023 dollar per ton of water. The system was an economic and efficient river water purification technology.

(2) The temperature directly affected the operation and treatment of the sequential series system of the constructed wetlands. The system was in a stage of operation and growth in April and May,

while the removal efficiency of pollutants was relatively low, and then gradually increases over time. The system was in a stable stage from June to September; thus, the removal efficiency of pollutants was the highest, and both removal rates of COD and $NH_4^+$-N were over 80%, while the TN exceeded 75%. The removal efficiency of pollutants decreased with the temperatures in the October. Then, the system stopped running when the water froze. In a temperate continental monsoon climate zone, the system cannot run stably all year; thus, we should take the measures of heat preservation and regulation of cold-adapted microorganism to prolong the operation time and improve purification efficiency.

(3) The sequential wetland system of FBCW, HSFCW, and SFCW by using the nearshore land on both sides of the river is an effective measure to treatment of polluted river water. The system can give full play to the advantages of different wetland and realize the maximization of the overall function. The system has high pollutant purification efficiency, good operation stability, strong anti-pollution load capacity, lower construction investment and operation cost, and simple management and maintenance. The technology can provide a reference for the restoration of polluted water bodies in urban rivers in temperate continental monsoon climate zone.

(4) The system integrates water purification, ecological restoration, landscape beautification, and entertainment. Based on the concept of "close to nature, multi-functional and sustainable" in the ecological restoration, the research gives a great theory and practice to promote the health of the river environment and the sustainable development of regional ecology economy society.

**Author Contributions:** Conceptualization, L.S.; methodology, L.S.; software, H.J.; validation, X.B., X.Z. and C.H.; formal analysis, X.B.; investigation, H.J.; resources, X.B.; data curation, X.B.; writing—original draft preparation, X.B.; writing—review and editing, X.B. and X.Z.; visualization, L.S. and J.Z.; supervision, L.S. and C.H.; project administration, X.B. and Z.W.; funding acquisition, C.H. and H.J. All authors have read and agreed to the published version of the manuscript.

**Funding:** This research was founded by the Foundation of Jilin Scientific and Technological Development Project, grant numbers (20190103137JH; 20190701048GH); the National Natural Science Foundation of China, grant numbers: 41901116.

**Conflicts of Interest:** The authors declare no conflicts of interest.

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
