# Peer review of "Purification Effect of Sequential Constructed Wetland for the Polluted Water in Urban River"

_water, doi:10.3390/w12041054_

Round 1

Reviewer 1 Report

The topic of this manuscript is of great interest. The rapid development of the urban areas caused by the industrialization and urbanization happened in the last century, have gradually deteriorated the water quality of the urban rivers.

The river water purification technology proposed in this paper based on a combination of different types of constructed wetlands, is very interesting thanks to the low costs, the ease of maintenance and management, the absence of secondary pollution, and the possibility of redeveloping the river banks within urban areas.

The manuscript is quite clearly written with an adequate bibliography so the proposed paper is suitable for publication with same revisions.

Same recommendation are reported below.

At lines 133-135: Make the format uniform in the text.

At line 143: Erase the space in the word particle.

At line 155: Check the number of figure.

Change Figure 3 with Figure 2.

At lines 193-194: Erase the sentence “The DO concentration of the outflow was 3.65-5.58 mg/L.”

I think this sentence is a typo from the previous paragraph.

At lines 195-197: Check the value of COD removal rate.

The authors wrote that the range of COD removal rate was 38.14%-82.05% but when they wrote the lowest value recorded in April the value is 38.23%.

At line 233: Check the value of TP concentration in the inflow.

The authors wrote that, during the operation of the system, the highest value of TP concentration was 2.69 mg/L but this value is not reported in the Figure 8 where all the values are close to 1.5 mg/L.

At lines 258-259: Check the number of figure.

The authors refer to the figures 10-12 that do not exist.

Author Response

We thank you for read and thoughtful comments on previous draft. We have carefully taken your comments into consideration in preparing our revision, which has resulted in paper that is clearer.

The following is a point-to-point response to the two reviewers’ comments.

Question: At lines 133-135: Make the format uniform in the text.

Response: We unified the format.

Question: At line 143: Erase the space in the word particle.

Response: We erased the space.

Question: At line 155: Check the number of figure.

Response: We checked the number of figure and changed to Figure 2.

Question: At lines 193-194: Erase the sentence “The DO concentration of the outflow was 3.65-5.58 mg/L.”

Response: We deleted the sentence “The DO concentration of the outflow was 3.65-5.58 mg/L.”

Question: At lines 195-197: Check the value of COD removal rate.

Response: We checked the raw date and analyzed the result. Then corrected the rate in the manuscript.

Question: At line 233: Check the value of TP concentration in the inflow.

Response: We checked the raw date and the concentration of TP in the inflow was 1.48-1.69 mg/L.

Question: At lines 258-259: Check the number of figure

Response: We checked and corrected for “figure 5-7”.

Reviewer 2 Report

Wetlands are the technologies known for many years. In literature exist many papers about the effectiveness of different system of wetlands in the removal of organic compounds and nitrogen and phosphorus compounds. Plaese add the novelty of your own research.
Please explain why the effectiveness of wetland system in summer were analyzed. What about the effectiveness of this system in other months when the temerature are lower than in the term June -September - please add the comment.

Author Response

We thank you for read and thoughtful comments on previous draft. We have carefully taken your comments into consideration in preparing our revision, which has resulted in paper that is clearer.

Question: Wetlands are the technologies known for many years. In literature exist many papers about the effectiveness of different system of wetlands in the removal of organic compounds and nitrogen and phosphorus compounds. Please add the novelty of your own research.  Please explain why the effectiveness of wetland system in summer were analyzed. What about the effectiveness of this system in other months when the temperature are lower than in the term June -September - please add the comment.

Response: In the Changchun, Northeast China, between June to September, the temperature gradually rose and the average temperature was 16-29℃, when the plants grew up. As the operating time increased, the operation of the system gradually became stable and the microbial activities increased significantly. The integrated effect of the wetland plants, substrate, and microorganisms was fully functional, so the purification efficiency reached the best.

We have added the more details of novelty of own research, reason and comment in the discussion.

Reviewer 3 Report

In this paper a sequential system of constructed wetland was built and used to treat the contaminated water of an urban river. The manuscript is well written, the methods are well described and the results are clearly presented. However, the paper is not innovative. Indeed, it simply reports the results of the water quality monitoring. The paper does not give any significant contribution for  the development of constructed wetland technologies. Furthermore, the discussion of results is unsatisfactory. In general the scientific merit is very low. For these reasons, in my opinion, the paper is not suitable for publication.

Author Response

We thank you for read and thoughtful comments on previous draft. We have carefully taken your comments into consideration in preparing our revision, which has resulted in paper that is clearer.

Question: In this paper a sequential system of constructed wetland was built and used to treat the contaminated water of an urban river. The manuscript is well written, the methods are well described and the results are clearly presented. However, the paper is not innovative. Indeed, it simply reports the results of the water quality monitoring. The paper does not give any significant contribution for the development of constructed wetland technologies. Furthermore, the discussion of results is unsatisfactory. In general the scientific merit is very low. For these reasons, in my opinion, the paper is not suitable for publication.

Response: The paper focused on the effect of the sequential system of constructed wetland, and there were no reports of the system in the practical engineering application.

We analyzed the results of the pollutant removal in the system and the single wetland in a year. Especially, the purification efficiency reached the best between June to September. The both removal rates of COD and NH4+-N were over 80%, when the TN could exceed 75%. We found the system was a good solution to the water purification of polluted river in cold zone. We hoped to provide a reference for the construction and operation of constructed wetlands in the cold zone. We also added some scientific discussion and conclusion for promotion of scientific merit.

Round 2

Reviewer 3 Report

The revision made by the authors did not improve the quality of the paper. Indeed, they simply included in the text some data already reported in the figures. My opinion on the scientific merit of the work remains unchanged. Therefore, the manuscript is not suitable for publication